

# Habitat and landscape factors influence pollinators in a tropical megacity, Bangkok, Thailand

Alyssa B. Stewart[1], Tuanjit Sritongchuay[2], Piyakarn Teartisup[3], Sakonwan Kaewsomboon[1] and Sara Bumrungsri[4]

[1] Department of Plant Science, Faculty of Science, Mahidol University, Bangkok, Thailand
[2] Center for Integrative Conservation, Xishuangbanna Tropical Botanical Garden, Chinese Academy of Sciences, Meglun, Mengla, Yunnan, China
[3] Faculty of Environment and Resource Studies, Mahidol University, Salaya, Nakhon Pathom, Thailand
[4] Department of Biology, Faculty of Science, Prince of Songkla University, Hat Yai, Songkhla, Thailand

## ABSTRACT

**Background:** Pollinators are well known for the ecosystem services they provide, and while urban areas are generally perceived as low-quality habitat for most wildlife, these cities often support a surprising degree of pollinator diversity. The current rapid growth of urban areas and concern over global pollinator declines have spurred numerous studies examining pollinator communities in temperate cities, but knowledge about tropical urban pollinators remains scarce.

**Methods:** This study investigated the effects of habitat and landscape factors on pollinator richness and abundance in a highly-populated, tropical city: Bangkok, Thailand. We conducted pollinator observations in 52 green areas throughout the city and collected data on patch size, floral abundance, plant richness, location type, and percent vegetation at five spatial scales.

**Results:** Of the 18,793 pollinators observed, over 98% were bees. Both patch size and floral abundance generally had positive effects on pollinators, although there was a significant interaction between the two factors; these findings were generally consistent across all focal taxa (*Tetragonula* stingless bees, *Apis* honey bees, *Xylocopa* carpenter bees, and butterflies).

**Discussion:** Our results demonstrate the importance of maintaining large green areas in cities, since small green areas supported few pollinators, even when floral resources were abundant. Moreover, most pollinator taxa utilized a variety of location types (e.g., public parks, school campuses, temple grounds), with the exception of butterflies, which preferred parks. Our findings are generally consistent with those of temperate urban studies, but additional studies in the tropics are needed before global patterns can be assessed.

Corresponding author
Alyssa B. Stewart,
alyssa.ste@mahidol.edu

# INTRODUCTION

The global human population size continues to increase, and much of this growth is concentrated in cities (*Grimm et al., 2008*; *Seto, Güneralp & Hutyra, 2012*). Such growth
has sparked a greater interest in urban ecology; a Web of Science search (conducted May 2018) for the terms "urban*" and "ecolog*" revealed 1,915 results from 2003–2007, 4,011 results from 2008–2012, and 8,305 results from 2013–2017. Thus, the literature in this field has doubled approximately every 5 years. This growing interest has stemmed, in part, as ecologists have come to recognize: (1) the large impacts of urban areas on local biodiversity, (2) the importance of preserving biodiversity in cities, and (3) the value of cities in modelling future biodiversity patterns, as urban areas are predicted to continue expanding worldwide (*Grimm et al., 2008*; *Bates et al., 2011*; *Robinson & Lundholm, 2012*; *Seto, Güneralp & Hutyra, 2012*).

One area of urban ecology that has received considerable attention is that of pollinator communities in cities. This interest is driven by a combination of general appreciation for the importance of pollinators (*Klein et al., 2007*; *Ollerton, Winfree & Tarrant, 2011*), and growing concern over global pollinator declines (*Potts et al., 2010* and references therein). Urbanization is generally predicted to have negative effects on pollinators (*Cariveau & Winfree, 2015*), with studies demonstrating greater habitat isolation in cities (*Ferreira, Boscolo & Viana, 2013*), reduced pollinator movement (*Bhattacharya, Primack & Gerwein, 2003*), and consequent reproductive isolation and higher rates of selfing in plants (*Harrison & Winfree, 2015*). Yet cities are often centers of high plant diversity and floral abundance (due to the cultivation of many exotic flowering plant species), which can lead to plentiful food and nesting resources (*McFrederick & LeBuhn, 2006*) and sometimes a surprising degree of pollinator diversity (*Hennig & Ghazoul, 2011*; *Hall et al., 2016*).

Consequently, many studies have investigated how various habitat and landscape factors influence pollinator communities. Some variables appear to be consistently important across diverse countries and pollinator taxa, such as floral abundance (*Ahrné, Bengtsson & Elmqvist, 2009*; *Bates et al., 2011*; *Hennig & Ghazoul, 2011*) and plant diversity (*Kearns & Oliveras, 2009*; *Bates et al., 2011*; *Hennig & Ghazoul, 2011*; *Hülsmann et al., 2015*). Other variables have received mixed support (such as patch size—*McFrederick & LeBuhn, 2006*; *Ahrné, Bengtsson & Elmqvist, 2009*; *Hennig & Ghazoul, 2011*; *Pardee & Philpott, 2014*), or appear to be important for only specific pollinator taxa (such as the amount of woody vegetation—*Pardee & Philpott, 2014*). Overall, the results seem to indicate there is some variation across locations and pollinator taxa, yet several patterns appear to be fairly universal.

However, before we can assess global patterns of pollinator responses to urbanization, more work must be conducted in tropical regions. Studies in temperate areas (*Biesmeijer et al., 2006*; *McFrederick & LeBuhn, 2006*; *Ahrné, Bengtsson & Elmqvist, 2009*; *Frankie et al., 2009*; *Kearns & Oliveras, 2009*; *Matteson & Langellotto, 2009*, *2011*; *Bates et al., 2011*; *Hennig & Ghazoul, 2011*; *Hanley, Awbi & Franco, 2014*; *Lowenstein et al., 2014*; *Pardee & Philpott, 2014*; *Baldock et al., 2015*; *Hülsmann et al., 2015*; *Banaszak-Cibicka, Ratyńska & Dylewski, 2016*; *Plascencia & Philpott, 2017*) have far outpaced studies in the tropics (*Frankie et al., 2013*), and we still know very little about the impacts of urbanization on tropical pollinator species. The two regions differ in a number of key attributes, such as the much greater biodiversity found in the tropics (*Gaston, 2000*;
*Willig, Kaufman & Stevens, 2003*). Moreover, tropical urban pollinators may face different challenges than temperate urban pollinators, given that they remain active year-round (*Bawa, 1990*). Previous studies have stressed the importance of pollination research in tropical cities (*Vanbergen & Insect Pollinators Initiative, 2013*; *Harrison & Winfree, 2015*), particularly as most of the world's future population growth will occur in the tropics (*Seto, Güneralp & Hutyra, 2012*). Thus, the objective of this research was to examine the effect of key habitat factors (patch size, floral abundance, plant richness, and location type) and landscape factors (percent vegetation within 100, 350, 650, 1,050, and 1,550 m buffers) on the pollinator community within a tropical megacity: Bangkok, Thailand.

## METHODS

### Study area

Data were collected in Bangkok, Thailand from January through April 2017. Bangkok is by far the largest urban area in Thailand, with over 9.6 million residents inhabiting 2,100 km$^2$ according to the most recent census in 2010 (*World Bank, 2015*). The landscape is dominated by man-made structures (business, residential, and transportation), and most of the vegetation found in Bangkok is intentionally cultivated and regularly managed. We selected 52 "green areas" (Table S1), defined as any location where vegetation comprises at least 50% of the ground area. We attempted to choose green areas evenly spaced throughout the city center (Fig. S1A), and these ranged in size from 600 m$^2$ to 1,140,000 m$^2$ (Table S1). We focused on the four most common types of green area: public parks, schools, temples, and commercial areas (Table S1).

### Pollinator observations

Within each green area, we conducted pollinator observations in 2 × 2 m plots spaced throughout the green area. Each plot was only observed once during the study period. Observations were conducted between 06:00 and 12:00 h; in a separate study, we found that pollinator richness and abundance did not differ significantly across time for 11 parks examined during both the morning (06:00–12:00 h, $n = 119$ plots) and afternoon (12:00–18:00 h, $n = 102$ plots) (*t*-test; richness: $t = 0.24$, $P = 0.8$; abundance: $t = 0.12$, $P = 0.9$). The number of plots per green area (Table S1) was approximately proportional to the size of the green area (more plots were conducted in larger green areas; *Ahrné, Bengtsson & Elmqvist, 2009*). In general, all plots within a green area were sampled on the same day, except for green areas with more than 20 plots, which were conducted over two consecutive days. To maximize the number of floral visitors encountered, we centered plots in floral-rich areas, and sampled the flowers of as many plant species as possible. Each 2 × 2 m plot was observed for 5 min, and if any visitors were observed during this period, we continued observations for an additional 10 min. All animals visiting during the 15-min observation period were recorded. We also noted whether or not visitors contacted floral reproductive structures; visitors that contacted stigmas and anthers were considered potential pollinators. Visitors that did not contact floral reproductive structures were not included in the analyses. When possible, pollinators were identified to species, but some taxa (e.g., *Tetragonula*, *Lasioglossum*) were difficult to

identify to species in the field (and even in the lab), and were therefore identified to genus or family. Unknown pollinators were either collected with a swing net or photographed, and then identified with the help of entomologists (see Acknowledgements). Permission to work with animals was granted by MUSC-IACUC (Faculty of Science, Mahidol University—Institute Animal Care and Use Committee) (License number MUSC60-038-388).

Pollinator richness was measured as the number of pollinator taxa per observation period, and pollinator abundance was quantified as the number of pollinator individuals per observation period. The abundances of focal pollinator groups (stingless bees, honey bees, carpenter bees, and butterflies) were also determined.

## Habitat and landscape factors

We collected data on four habitat factors: patch size, location type, floral abundance, and plant richness. Patch size and location type were measured at the level of the green area, while floral abundance and plant richness were measured at the level of the plot. Patch size, or the total expanse of land within each green area (not including the area covered by large bodies of water), was measured using the "satellite" view in Google Maps. Each green area was also categorized into one of four location types: public parks, schools, temples, and commercial areas. We recorded both floral abundance (the number of flowers within each plot) and the floral abundance of attractive plants (defined as plant species whose flowers were visited by at least one pollinator). Because Bangkok has many cultivated plant species that are unattractive to pollinators (Table S2), attractive floral abundance was considered more appropriate for addressing our research objectives than total floral abundance. Indeed, compared to total floral abundance, attractive floral abundance was found to be a better predictor of both pollinator richness (Pearson correlation: $r = 0.363$ versus $r = 0.326$) and pollinator abundance (Pearson correlation: $r = 0.244$ versus $r = 0.204$). We therefore used the floral abundance of attractive plants for all analyses (hereafter referred to as simply "floral abundance"). Plant richness was quantified as the total number of plant species observed at all plots within a green area (a proxy for total plant richness of the green area) divided by the number of plots conducted (to account for the fact that more plots were observed in larger green areas). Plants were identified to species or genus, not to the level of cultivar.

For landscape factors, we calculated the percentage of vegetated area surrounding each plot within 100, 350, 650, 1,050, and 1,550 m concentric circles (Fig. S1B). We used the following secondary data sources: (a) Bangkok Metropolis map (1:4,000 scale) and (b) IKONOS 2017 satellite images at one m resolution. The satellite image was used to classify vegetated and non-vegetated areas, and converted into digital format through supervised classification in ArcGIS 10.4 software. We then used ArcCatalog to create a new geodatabase (containing spatial attributes) and data set (containing non-spatial attributes) for each spatial scale (100, 350, 650, 1,050, and 1,550 m radiuses). The output was then used to calculate percent vegetation (vegetated area divided by the sum of vegetated and non-vegetated areas) surrounding each plot at each spatial scale.

## Statistical analysis

All analyses were conducted in R 3.4.4 (*R Core Team, 2018*). We used generalized linear mixed modeling to determine the effect of focal predictors on the following response variables: total pollinator richness, total pollinator abundance, and the abundance of each focal pollinator group (stingless bees, honey bees, carpenter bees, and butterflies). Green area was included as a random factor (as multiple plots were conducted per green area) and the Poisson distribution was used for all models (as all response variables were counts).

Model selection followed modified methods of *Bates et al. (2011)*. For each spatial scale (0, 100, 350, 650, 1,050, and 1,550 m buffers), nested likelihood ratio tests were used to determine the significance of each habitat factor of interest: patch size, location type, floral abundance, plant richness, the interaction between patch size and floral abundance, and the interaction between patch size and plant richness. We then selected the top model from each spatial scale and compared their AIC (Akaike information criterion) values to select the best model overall (the model with the lowest AIC).

To visualize plant-pollinator interactions as a whole, and to analyze pollinator diet breadth, we constructed and analyzed a pollination network using the "bipartite" package (*Dormann, Gruber & Fruend, 2008*; *Dormann et al., 2009*) in R 3.4.4. We pooled the interaction data of all 52 green areas and used the function "plotweb" to build the network. We then used the function "specieslevel" to examine diet generalization for each pollinator taxa with the metrics "degree" and "normalized degree." Degree refers to the number of floral hosts per pollinator taxa. Normalized degree is calculated as degree divided by the number of possible interacting partners, which accounts for differences in network size. Normalized degree is a quantitative measure of a pollinator species' position in the network; species with high normalized degrees are central to network structure and promote network robustness (*Sole & Montoya, 2001*; *Dunne, Williams & Martinez, 2002*). Moreover, high normalized degrees (near 1) indicate more generalized diets, while small degrees (near 0) indicate greater specialization.

# RESULTS

## Plant species

A total of 140 flowering plant taxa (46 families; Table S2) were observed from 469 plots in the 52 green areas. The most specious plant families observed in this study were Leguminosae (19 species), Apocynaceae (11 species), and Acanthaceae (nine species). Plots had an average of 221.2 ± 27.4 flowers (range: 1–9,000 flowers per plot). The three most commonly observed plant species were *Ruellia simplex* C. Wright (observed in 46 plots), *Cassia fistula* L. (32 plots), and *Caesalpinia pulcherrima* (L.) Sw. (28 plots). The plant species that attracted the most pollinator individuals per flower (number of pollinators observed divided by number of flowers recorded in single species plots) were *Gustavia gracillima* Miers (22.2 pollinators per flower), *Nymphaea nouchali* Burm.f. (10.1 pollinators per flower), and *Nelumbo nucifera* Gaertn. (9.75 pollinators per flower). The plant species that attracted the greatest diversity of pollinator species were *Citrus maxima* (Burm.) Merr.
**Table 1 Results of the generalized linear mixed models (GLMM) for our six response variables.**

| | Total pollinator richness | Total pollinator abundance | Stingless bee abundance | Honey bee abundance | Carpenter bee abundance | Butterfly abundance |
|---|---|---|---|---|---|---|
| **Habitat** | | | | | | |
| Patch size × Floral abundance | $\chi^2_1 = 5.48$ **$P = 0.019$** | $\chi^2_1 = 1,546$ **$P < 0.001$** | $\chi^2_1 = 956$ **$P < 0.001$** | $\chi^2_1 = 530$ **$P < 0.001$** | $\chi^2_1 = 81.3$ **$P < 0.001$** | $\chi^2_1 = 83.0$ **$P < 0.001$** |
| Patch size × Plant richness | $\chi^2_1 = 0.98$ $P = 0.322$ | $\chi^2_1 = 1.20$ $P = 0.273$ | $\chi^2_1 = 0.60$ $P = 0.440$ | $\chi^2_1 = 0.19$ $P = 0.660$ | $\chi^2_1 = 0.29$ $P = 0.589$ | $\chi^2_1 = 0.98$ $P = 0.322$ |
| Patch size | $\chi^2_1 = 10.1$ **$P = 0.002$** | $\chi^2_1 = 6.04$ **$P = 0.014$** | $\chi^2_1 = 2.74$ $P = 0.098$ | $\chi^2_1 = 7.82$ **$P = 0.005$** | $\chi^2_1 = 4.38$ **$P = 0.036$** | $\chi^2_1 = 10.9$ **$P < 0.001$** |
| Floral abundance | $\chi^2_1 = 23.6$ **$P < 0.001$** | $\chi^2_1 = 2,576$ **$P < 0.001$** | $\chi^2_1 = 1,506$ **$P < 0.001$** | $\chi^2_1 = 641$ **$P < 0.001$** | $\chi^2_1 = 122$ **$P < 0.001$** | $\chi^2_1 = 139$ **$P < 0.001$** |
| Plant richness | $\chi^2_1 = 0.29$ $P = 0.593$ | $\chi^2_1 = 0.12$ $P = 0.729$ | $\chi^2_1 = 0.14$ $P = 0.711$ | $\chi^2_1 = 0.21$ $P = 0.649$ | $\chi^2_1 = 2.87$ $P = 0.090$ | $\chi^2_1 = 3.41$ $P = 0.065$ |
| Location type | $\chi^2_3 = 3.29$ $P = 0.349$ | $\chi^2_3 = 1.44$ $P = 0.697$ | $\chi^2_3 = 0.24$ $P = 0.971$ | $\chi^2_3 = 1.85$ $P = 0.603$ | $\chi^2_3 = 1.45$ $P = 0.695$ | $\chi^2_3 = 13.2$ **$P = 0.004$** |
| **Landscape (AIC)** | | | | | | |
| None | 1,419 | 19,029 | 12,801 | 13,822 | 1,238 | 658 |
| 100 m | 1,411 | 18,686 | **12,511** | 13,713 | 1,172 | 651 |
| 350 m | 1,410 | **18,511** | 12,583 | 13,448 | **1,167** | **647** |
| 650 m | **1,409** | 18,561 | 12,677 | **13,315** | 1,206 | 654 |
| 1,050 m | 1,410 | 18,807 | 12,753 | 13,504 | 1,230 | 657 |
| 1,550 m | 1,412 | 18,944 | 12,773 | 13,690 | 1,234 | 655 |

**Note:**
The response variables were: total pollinator richness, total pollinator abundance, and the abundances of four focal taxa (*Tetragonula* stingless bees, *Apis* honey bees, *Xylocopa* carpenter bees, and butterflies). The significance of the six habitat predictors (four main factors and two interactions) were tested using nested likelihood ratio tests. The landscape scale (surrounding percent vegetation at 0, 100, 350, 650, 1,050, and 1,550 m buffers) that best fit each model was determined from AIC values. The significant factors that were included in the final models are highlighted in yellow (with P-value or AIC value bolded).

(five pollinator species), *Justicia betonica* L. (five species), *Buddleja paniculata* Wall. (4.5 species), *Peltophorum pterocarpum* (DC.) K. Heyne (4.2 species), and *Syzygium jambos* (L.) Alston (four species) (See Table S2 for more detailed results).

## Pollinator species

Throughout this study, we observed a total of 18,793 pollinators comprising 40 taxa from six orders (Table S3). Hymenopterans (bees and wasps) were by far the most abundant, comprising 98.63% of all pollinators observed. Lepidoptera (butterflies and moths; 1.16%), Diptera (flies; 0.15%), Coleoptera (beetles; 0.032%), and Hemiptera (true bugs; 0.021%) were also observed contacting floral reproductive structures, as was one sunbird (order Passeriformes, family Nectariniidae; 0.005%). The most common pollinators were stingless bees (64.32%), of which only a single genus was observed (*Tetragonula*). Next most abundant were the three species of honey bees (*Apis florea*, 12.99%; *Apis cerana*, 12.48%; *Apis dorsata*, 6.11%) and one species of carpenter bee (*Xylocopa aestuans*, 1.45%). The abundances of the remaining pollinator taxa each accounted for less than 1% of total pollinator observations (See Table S3 for more detailed results).

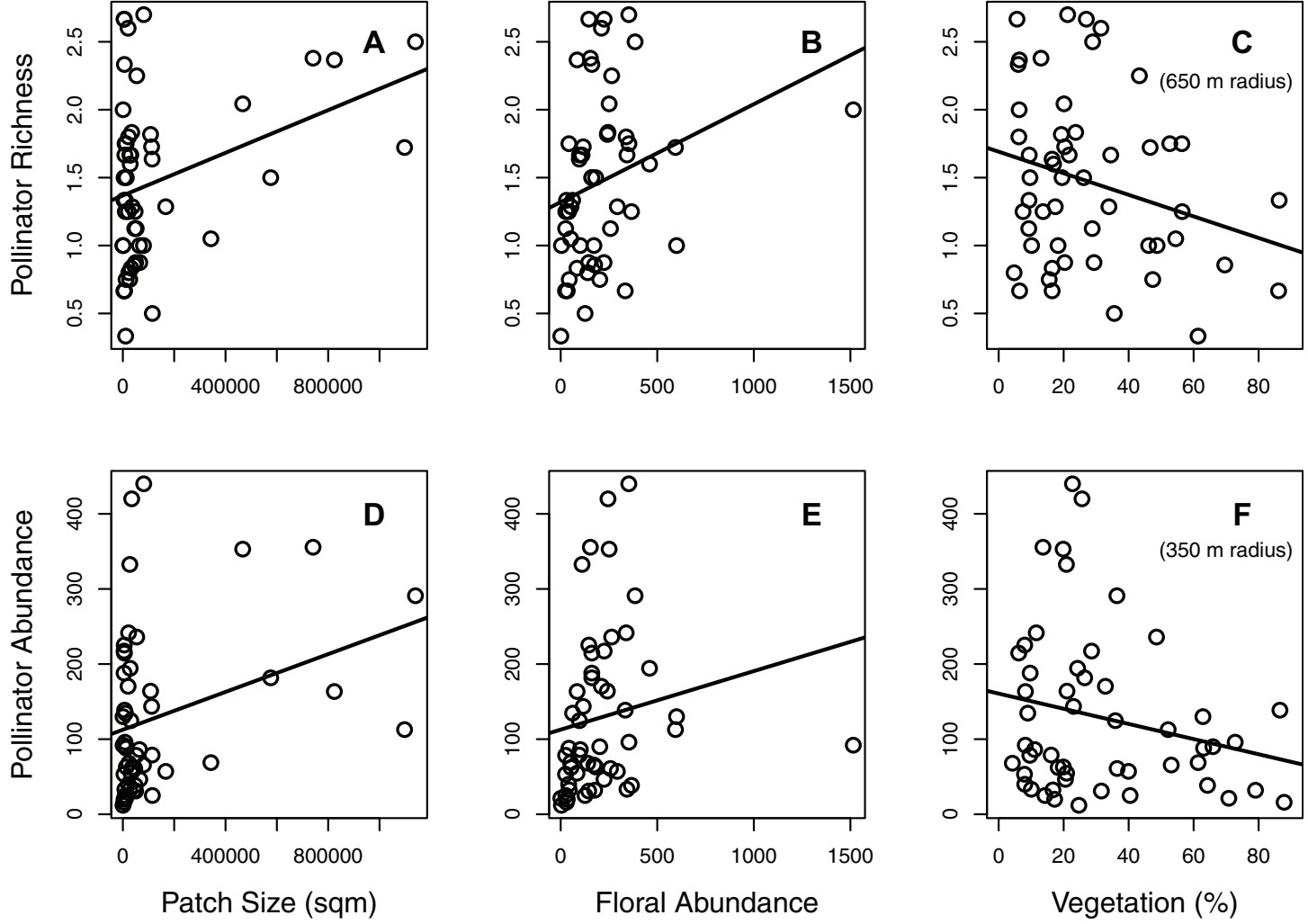

**Figure 1 Effects of significant habitat and landscape factors on total pollinator richness and abundance.** Two main factors (patch size and floral abundance) had significant positive effects on both richness (A–C; number of species per 15-min observation period) and abundance (D–F; number of individuals per hour), while one main factor (surrounding percent vegetation) had a significant negative effect. Surrounding percent vegetation at the 650 m radius best described pollinator richness, while the 350 m radius best described pollinator abundance. Each circle represents the mean of one green area (values averaged across all plots within the green area).

## Habitat and landscape factors

The effects of habitat and landscape factors were relatively similar for both pollinator richness and abundance (Table 1; Figs. 1 and 2). When combining the data of all pollinator taxa, both total pollinator richness and total pollinator abundance were significantly influenced by patch size, floral abundance, and their interaction (Table 1; Figs. 1 and 2). However, total pollinator richness was best described by the model including percent vegetation within a 650 m radius, while total pollinator abundance was best described by the model including percent vegetation within a 350 m radius (Table 1).

When examining the abundance of focal taxonomic groups, the results were generally consistent with those of total pollinator abundance (Table 1; Figs. S2A and S2C). The abundance of honey bees and carpenter bees were each significantly influenced by

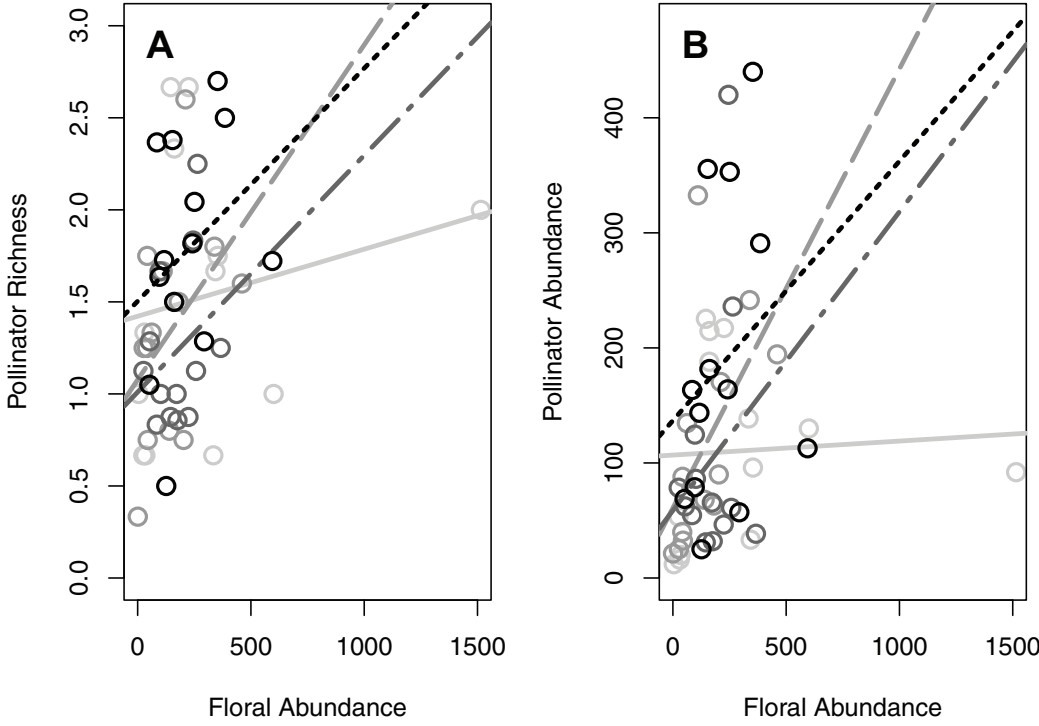

**Figure 2 The significant interaction between patch size and floral abundance.** Both (A) total pollinator richness (number of species per 15-min observation period) and (B) total pollinator abundance (number of individuals per hour) were influenced by a significant interaction between patch size and floral abundance. Each circle represents the mean of one green area (values averaged across all plots within the green area). The different colors and line types represent the different patch size classes: light grey (solid line) = small green areas (600–900 m²; n = 13), medium grey (dashed) = medium size green areas (901–30,000 m²; n = 13), dark grey (dot-dashed) = large green areas (30,001–81,000 m²; n = 13), black (dotted) = very large green areas (81,001–1,140,000 m²; n = 13).

the same three factors as overall abundance: patch size, floral abundance, and their interaction (Table 1; Figs. S2A and S2C). The abundance of butterflies was also affected by the same three factors, as well as location type (Table 1; Figs. S2A–S2C). Finally, the abundance of stingless bees was influenced by floral abundance and the interaction between patch size and floral abundance (but not patch size itself; Table 1; Figs. S2A and S2C). The results for percent vegetation were more variable; stingless bees were most influenced at the 100 m scale, butterflies and carpenter bees at the 350 m scale, and honey bees at the 650 m scale (Table 1).

Significant habitat and landscape factors nearly always resulted in the same pattern for the different taxonomic groups (Figs. 1 and 2; Figs. S2A and S2C). Larger patch size and greater floral abundance, when significant, increased pollinator richness and abundance (Fig. 1; Fig. S2A). On the other hand, greater percent vegetation generally reduced pollinator richness and abundance, although its effect on the abundance of carpenter bees and butterflies was minimal (Fig. 1; Fig. S2A). Regarding significant interactions between patch size and floral abundance, greater floral abundance increased both pollinator
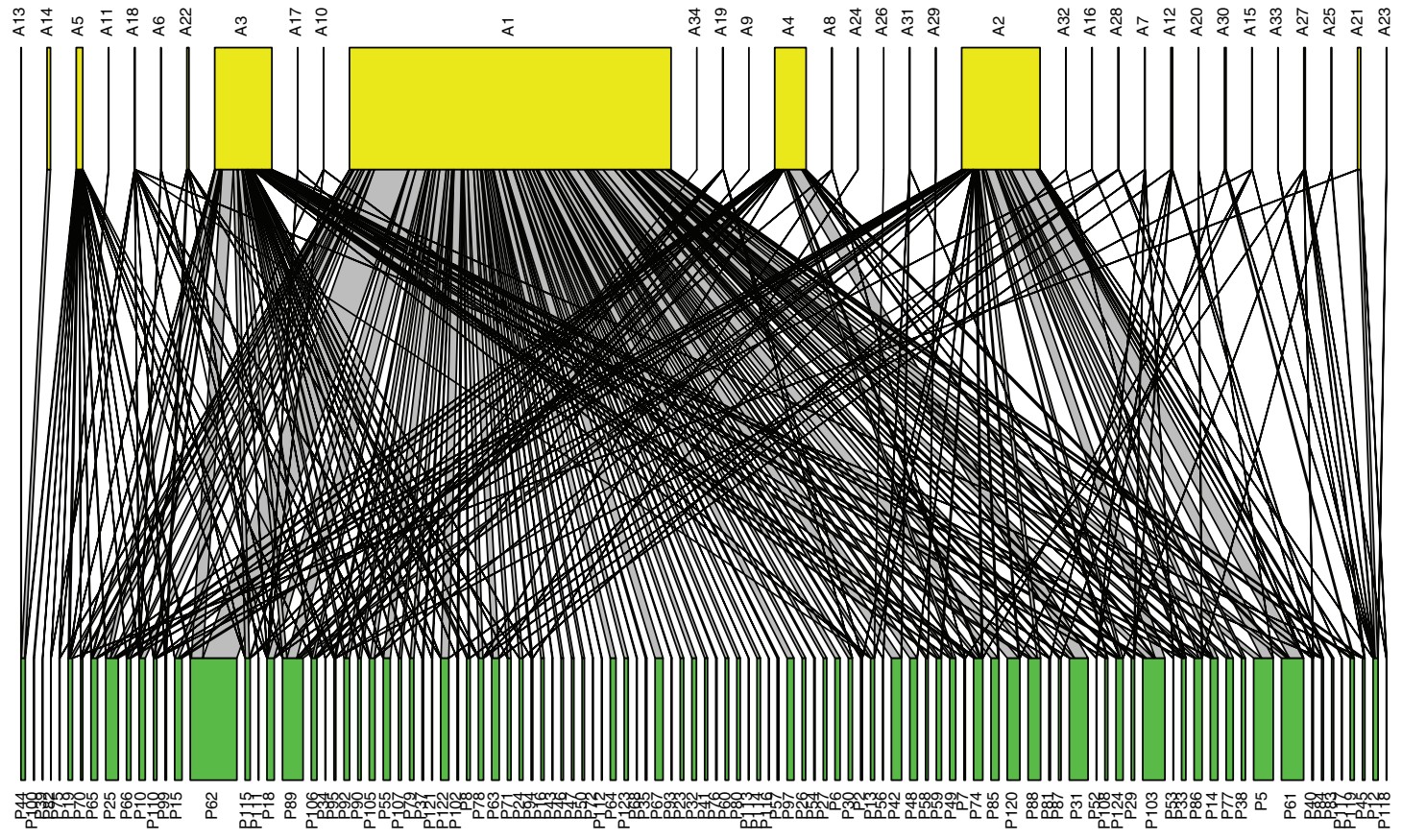

**Figure 3 Pollination network of plants and pollinators in Bangkok, Thailand.** Pollinator taxa are listed along the top (yellow), and plant taxa are listed along the bottom (green). The network was constructed from pollinator abundance data at each plant species; a line connecting pollinator species *i* to plant species *j* represents an observed interaction between the two, and the width of the connecting line is proportional to the average number of *i* pollinator individuals observed visiting plant species *j*. The five most abundant pollinator taxa were: A1 = *Tetragonula*, A2 = *Apis florea*, A3 = *A. cerana*, A4 = *A. dorsata*, and A5 = *Xylocopa aestuans* (a higher resolution figure is available in Fig. S3, and all species names are listed in Table S4).

richness and abundance in the larger patches, but did not increase these metrics in the smallest patches (Fig. 2; Fig. S2C).

## Pollination network

Our pollination network (Fig. 3; Fig. S3) revealed that *Tetragonula*, the three *Apis* species, and *X. aestuans* were not only the most common pollinators, but also visited a high diversity of plant species. Examining the network metrics confirmed that these taxa had the most generalized diets: *Tetragonula* visited 93 plant species, followed by *A. cerana* (42 species), *A. florea* (33 species), *X. aestuans* (19 species), and *A. dorsata* (17 species) (See Table S4 for more detailed results).

## DISCUSSION

### Pollinator abundance and richness

Pollinator abundance was surprisingly high, given the extent of urbanization in Bangkok. We observed an average of 40.1 ± 2.8 (mean ± SE, *n* = 469 plots) pollinators per 2 × 2 m

plot per 15-min observation period (range: 0–422 pollinators). In comparison, *Hennig & Ghazoul (2011)* reported an average of 32 visits per 2 × 2 m plot (observed for 40 min each). The high pollinator abundances observed in this study were due in large part to the *Tetragonula* stingless bees; it was not uncommon to observe between 100 and 300 stingless bees at plots with high abundances of attractive flowers. The three *Apis* species were occasionally observed in high numbers (up to around 100 individuals of a single *Apis* species), but it was extremely uncommon for any of the other taxa to exceed 10 individuals per observation period.

In contrast, we found lower pollinator richness in Bangkok (40 taxa) than has been reported in previous urban studies. *Hennig & Ghazoul (2011)* observed 148 insect species in Zürich, Switzerland, while *Baldock et al. (2015)* noted 147 visitor taxa at 12 urban sites throughout the UK. Moreover, reports of urban bee species richness alone have ranged between 37 and 64 species (*Kearns & Oliveras, 2009*; *Bates et al., 2011*; *Matteson, Ascher & Langellotto, 2008*; *Frankie et al., 2013*; *Lowenstein et al., 2014*). There are likely two factors contributing to the low richness observed in this study. Firstly, many common taxa could only be classified to genus or family in the field (e.g., *Tetragonula*, *Lasioglossum*, Lycaenidae; Table S3). More specific identification would undoubtedly raise our measures of species richness. Secondly, Bangkok's lower richness is not surprising given the extent of urbanization in the city. Most of the studies mentioned above were conducted in much smaller cities (less than three million residents), with the exception of the study by *Matteson, Ascher & Langellotto (2008)* that was conducted in New York City. Thus, we predict that only pollinator species unfazed by a highly anthropogenic environment can persist in Bangkok.

The most common species observed in Bangkok do appear to be species with generalist diets and adaptable nesting behaviors. The reason we suggest a generalist diet is important is because the floral environment is almost always highly managed and frequently altered. Yet regularly switching blooming plant species may actually benefit generalist, urban pollinators by providing them with a continuous supply of floral resources. This constant floral supply may be particularly beneficial in tropical areas such as Bangkok, where pollinator species forage year-round, unlike temperate pollinators which are only active during the warmer months (*Bawa, 1990*). The pollination network constructed from our data reveals that the most common insect species do indeed have broad diets. Moreover, all of the common bee species observed in our study appear to have adaptable nesting behaviors. At one public park, we observed a *Tetragonula* colony that had made their home in a small, plastic electrical box. Previous work has also reported stingless bees (*Heard, 1999*), *A. cerana* (*Inoue, Adri & Salmah, 1990*), *A. dorsata* (*Deodikar et al., 1977*; *Nagaraja & Yathisha, 2015*), *A. florea* (*Wongsiri et al., 1997*), and *Xylocopa* bees (*Watmough, 1973*) utilizing man-made structures for nesting.

## Effects of patch size, floral abundance, and plant richness

The two main effects that consistently had a positive influence on the pollinator indices examined were patch size and floral abundance. These findings are generally consistent with the results of previous studies. Floral abundance appears to be the most universally positive factor, significantly influencing bees (*McFrederick & LeBuhn, 2006*; *Ahrné,*

*Bengtsson & Elmqvist, 2009*; *Matteson & Langellotto, 2010*; *Bates et al., 2011*; *Hennig & Ghazoul, 2011*; *Pardee & Philpott, 2014*), hoverflies (*Bates et al., 2011*; *Hennig & Ghazoul, 2011*), and butterflies (*Hardy & Dennis, 1999*; *Matteson & Langellotto, 2010*). Patch size is less consistent; some studies have found that measures of pollinator richness or abundance increase with larger patches (*Hennig & Ghazoul, 2011*; *Pardee & Philpott, 2014*), while others have reported that patch size has no effect on pollinators (*McFrederick & LeBuhn, 2006*; *Ahrné, Bengtsson & Elmqvist, 2009*). Findings for plant richness have also been mixed, with some studies reporting minimal effects (*Banaszak-Cibicka, Ratyńska & Dylewski, 2016*; *Plascencia & Philpott, 2017*), as in our study, and others reporting a significant positive effect (*Kearns & Oliveras, 2009*; *Bates et al., 2011*; *Hennig & Ghazoul, 2011*; *Hülsmann et al., 2015*). In Bangkok, it appears that floral abundance is more important that plant richness, which may be explained by the apparent flexible diets of most observed pollinator species. Moreover, the importance of floral abundance and patch size likely results from the fact that the former is a direct measure of food availability and the latter is possibly correlated with nest site availability, both of which are important to pollinators (*Baldock et al., 2015*).

## Interaction between patch size and floral abundance

Interestingly, our results also revealed a consistent significant interaction between patch size and floral abundance. The interaction revealed, for the larger green areas, a strong positive correlation between floral abundance and all pollinator indices. However, for the smallest green areas, increasing floral abundance did not greatly increase pollinator richness or abundance. We suggest three potential explanations for these patterns. Firstly, since pollinator richness increases with floral abundance in the large areas but not the small areas, we suggest that small green areas lack the diversity of nesting sites needed to recruit greater pollinator richness. Secondly, since the same pattern occurs for pollinator abundance, we suggest that pollinators do not become more abundant with increasing floral abundance in the smallest green areas due to limited nesting availability in these areas. Finally, it is possible that some pollinator species prefer larger foraging grounds than what is offered by small green areas (*Pauw, 2007*). Previous studies have not looked for an interaction between patch size and floral abundance, and further studies are required to determine if such patterns are common in other urban areas.

## Effect of percent vegetation

The spatial scale that best fit our pollinator richness and abundance data differed by taxa, which likely reflects their different foraging ranges. *Tetragonula* stingless bees (best described by percent vegetation within 100 m) are relatively small pollinators with correspondingly small foraging distances (120–850 m; *Van Nieuwstadt & Iraheta, 1996*; *Smith et al., 2017*). *Kuhn-Neto et al. (2009),* however, did report that the stingless bee *Melipona mandacaia* could forage up to 2.1 km. *Xylocopa* carpenter bees and butterflies (best described at the 350 m scale) have been reported to have longer foraging distances. For example, *Pasquet et al. (2008)* found that *X. flavorufa* could forage between 50 and 6,040 m (median = 720 m), *Cant et al. (2005)* observed that the style of butterfly flight
characterized by foraging activity spanned 1,210.9 ± 455.4 m, and *Ovaskainen et al. (2008)* reported that *Melitaea cinxia* butterflies flew between 9 and 5,490 m (median = 505 m). *Apis* honey bees (best described at the 650 m scale) have even longer maximum foraging distances: at least one km, and potentially up to 12 km (*Dyer & Seeley, 1991*). While the spatial scales that best fit our models are smaller than previously reported foraging distances, it is consistent with prior work which has suggested that pollinators in urban areas may have smaller foraging ranges than those in natural habitats (*López-Uribe, Oi & Del Lama, 2008*).

The negative correlation between percent vegetation and our various pollinator indices was contrary to our initial hypothesis. Indeed, previous studies have generally found that greater vegetation in the surrounding landscape increases measures of pollinators or pollination success (*Bates et al., 2011*; *Sritongchuay, Kremen & Bumrungsri, 2016*). However, we suggest that the vegetation in Bangkok is inherently different than the natural, uncultivated vegetation encountered in previous studies. First of all, much of the vegetation in Bangkok consists of hedges and lawns, which often do not offer floral resources. Moreover, many species considered attractive to humans and planted in abundance throughout the city were observed to be highly unattractive to insects (e.g., *Bougainville*, *Plumeria*, *Hibiscus*; Table S2). Finally, much of Bangkok's vegetation is highly managed (hedges and trees are trimmed and lawns are mown), which may deter insect pollinators from utilizing them as either nesting or foraging sites. Similar to our results, *Banaszak-Cibicka, Ratyńska & Dylewski (2016)* also reported a negative correlation between urban bee density and shrub cover, and *Matteson & Langellotto (2010)* found no effect of landscape level vegetation on bee richness. While urban vegetation certainly provides other benefits (*Smith et al., 2006*; *Robinson & Lundholm, 2012*), it does not necessarily promote pollinator richness or abundance.

## Effect of location type

Most pollinator taxa did not appear to differentiate between the various location types (public parks, schools, temples, and commercial areas). The one exception was the butterflies, which were more common in public parks than the other three location types. Butterflies were the most skittish insect pollinators that we observed, so human density or activity may have reduced butterfly abundance at schools, temples, and commercial areas (although this is just speculation as we did not quantify human density or activity). However, all other pollinator groups were unaffected by location type. There are two key factors that likely contribute to these findings. The first is that Bangkok's schools, temples, and other commercial areas typically have higher floral abundances than the surrounding urban landscape, and share similar characteristics with public parks. Notably, in our study, there were no significant differences among the four location types for either floral abundance (ANOVA, $F = 1.39$, $P = 0.26$) or patch size (ANOVA, $F = 2.1$, $P = 0.11$). The second key factor is that the pollinator taxa observed in Bangkok are relatively generalist species, with flexible foraging and nesting requirements (as covered at the beginning of the Discussion). Our results indicate that school campuses, temples, and even commercial areas can be just as effective as public parks in promoting urban pollinator richness and abundance.

### Native versus exotic plant species

While not the focus of this study, we did also compare pollinator preferences for native ($n$ = 43 species) versus exotic ($n$ = 65 species) plants. Overall pollinator richness and abundance, as well as abundance of each of the focal taxa, were not significantly different between native and exotic plant species (Table S5). Previous studies have reported mixed findings. For example, Chrobock et al. (2013) found that pollinator visitation rates to native plant species were higher than those to exotic species, while Matteson & Langellotto (2011) reported that bumble bees and Apis mellifera visited native and exotic species equally, yet megachilid bees and butterflies actually favored exotic plant species. Moreover, Hanley, Awbi & Franco (2014) found that bumblebee species with specialized diets preferred native species and bumblebee species with generalized diets preferred exotic species, and Pardee & Philpott (2014) reported that native bee species preferred native gardens over non-native gardens. While all of the pollinator species in our study (that we had information for) were native (Table S3), their lack of preference between native and exotic plant species may be explained in part by their generalized diets. Moreover, given their prevalence in Bangkok, exotic plants may be important in sustaining the city's pollinator populations, as has been suggested for other urban areas (Shapiro, 2002; Matteson & Langellotto, 2011; Harrison & Winfree, 2015).

## CONCLUSIONS

The results of this study provide important information about pollinator communities in a highly-populated, tropical city, which are vastly understudied compared to their temperate counterparts. The habitat factors exerting the largest influence on pollinator richness and abundance were patch size and floral abundance, yet the significant interaction between these two factors revealed that the smallest green areas (ca. <10,000 m$^2$) retained low pollinator richness and abundance, even when floral abundance was very high. Such findings highlight the importance of retaining large green areas in cities. Yet this work also reveals that the meaning of the term "green area" is not necessarily restricted to traditional public parks, as other patch types with high floral abundance can support urban pollinators. We suggest additional work examining pollinator communities in large cities, particularly in the tropics, where most of the world's population growth is expected to occur.

## ACKNOWLEDGEMENTS

We thank Dr. Natapot Warrit and Dr. Tom Stewart for helping identify insect specimens and photos, and Dr. Prapeut Kerdsueb for helping with GIS analyses. We also thank Dr. Gail Langellotto and one anonymous reviewer for comments on an earlier draft.

### Funding

This research was supported by the Thailand Research Fund (grant MRG6080124 to Alyssa B. Stewart) and Mahidol University (Mentorship Grant co-awarded to

Alyssa B. Stewart). The funders had no role in study design, data collection and analysis, decision to publish, or preparation of the manuscript.

## Grant Disclosures
The following grant information was disclosed by the authors:
Thailand Research Fund: MRG6080124.
Mahidol University.

## Competing Interests
The authors declare that they have no competing interests.

## Author Contributions
- Alyssa B. Stewart conceived and designed the experiments, performed the experiments, analyzed the data, prepared figures and/or tables, authored or reviewed drafts of the paper, approved the final draft.
- Tuanjit Sritongchuay performed the experiments, analyzed the data, prepared figures and/or tables, approved the final draft.
- Piyakarn Teartisup analyzed the data, contributed reagents/materials/analysis tools, prepared figures and/or tables, approved the final draft.
- Sakonwan Kaewsomboon performed the experiments, approved the final draft.
- Sara Bumrungsri conceived and designed the experiments, approved the final draft.

## Animal Ethics
The following information was supplied relating to ethical approvals (i.e., approving body and any reference numbers):
Permission to work with animals was granted by MUSC-IACUC (Faculty of Science, Mahidol University–Institute Animal Care and Use Committee) (license number MUSC60-038-388).

## Data Availability
The raw data are provided as Supplemental Files.

## Supplemental Information
Supplemental information for this article can be found online at http://dx.doi.org/10.7717/peerj.5335#supplemental-information.

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
