# Peer review of "Habitat and landscape factors influence pollinators in a tropical megacity, Bangkok, Thailand"

_PeerJ, doi:10.7717/peerj.5335_

## Round 0.1 · original submission · Major Revisions

Please clearly address all of the reviewer comments in the manuscript and in a rebuttal letter. Thank you.

·

Basic reporting

The manuscript 'Habitat and landscape factors influence pollinators in a tropical megacity, Bangkok, Thailand', is an extremely well-written report of pollinator abundance and diversity, as well as the factors that promote or impeded abundance/diversity, in Bangkok, Thailand.

I have a few comments for improvement, related to references and one figure. These comments are relatively minor.

***Line 271: Lowenstein was conducted in Chicago, IL, which many would consider to be a large city.
***Line 268 and Line 275: Matteson and Langellotto (2011) is cited in these two lines. However, I believe the correct citation is Matteson et al. (2009). The 2009 paper is where we report on bee diversity in NYC gardens. Our 2011 paper secondarily cites our 2009 data.
***I hate to self-promote, but I do believe that Matteson and Langellotto (2010, Determinates of inner city butterfly and bee species richness, Urban Ecosystems 13(3): 333-347) would make a nice addition to this paper, in lines 296-297, and also in line 343. We used a different analysis (Bayesian analysis, rather than GLMM), but also found that garden size (patch size) and floral abundance best explained bee richness in NYC. We found that landscape level greenspace had no impact on bee richness). Since the results align with what the authors found in Bankok, Thailand, I thought it might make a good addition.
***In figure 2, is there any way to change the gray-scale lines to various dashed or dotted lines? I personally had a difficult time distinguishing between the two lightest scale grays.

Experimental design

In the manuscript 'Habitat and landscape factors influence pollinators in a tropical megacity, Bangkok, Thailand', the authors report the results of a four month study of pollinator abundance and diversity across 52 plots. Although the time scale for the study is fairly short term, the spatial extent of the study is impressive. The large number of pollinators observed over the course of the study (>18,000 pollinators) suggests that the authors sufficiently sampled the sites.

Although there are an increasing number of published studies on pollinator diversity from temperate cities, the authors correctly point out that the number of published studies on pollinator diversity from tropical cities is a void in current literature.

I have a few minor comments on the methods.

***Line 110: what methods were used to compare pollinator abundance and diversity between the two time periods? Specifically, how many patches were observed, and how many times?
****Line 127: pollinator diversity would be more accurately described as pollinator richness
****138-139: attractive plants are listed as plant species, rather than as ornamental cultivars. I would be surprised if wild type plants, rather than cultivars, were planted in these public spaces. If that is the case, but the authors identified plants to species, rather than cultivar, please say so. Or, if these were truly wild-type plants, not bred for ornamental displays, please say so.
****In the pollinator observation sections, could you please provide more details on the pollinator sampling? Were all plots sampled once between January and April 2017? Were plots sampled multiple times during this time period? Were all plots sampled every day or every week during this time period? It is hard for the reader to understand the sampling extent, without these additional details.

Validity of the findings

I have no comments for improvement on the data or conclusions.

Reviewer 2 ·

Basic reporting

1. Despite being an overall well-written article, I had an initial negative impression of the writing. This is likely because the very first sentence (lines 20-22) is the most poorly written sentence of the entire article. The sentence is both vague (e.g., urban area biodiversity is relatively limited compared to what?) and inaccurate (i.e., your results and all others cited show that a lot more than a "few" pollinator species are found in cities) and therefore, requires editing or rewriting. Lines 29-30 should also be made less vague, for example by stating the direction the most influential factors had (i.e., positive). A stronger and more clear abstract will give readers a better sense of the article's merit before diving in for the full read.
2. The authors have provided adequate introductory material/ context and have covered their bases in terms of relevant papers. However, a reference may be needed at lines 140-141. Have other studies excluded non-attractive plants from floral abundance measurements? If so, an example should be cited.
3. Line 225 remove the word smaller

Experimental design

A meaningful exploration of an important (and large) knowledge gap is addressed through this study of urban pollinators in a tropical city.
1. The overall study design appears sound, however, some vital details are missing. A time range is indicated for when the data were collected (Line 97), but no other information is provided about how sites were sampled over this period. E.g., was each plot only sampled once during this period or multiple times? Were all plots in a green area visited on the same day? Were multiple green areas visited on the same day? What was the rationale behind these sampling decisions? My main concern here is that some areas may lack continuous blooms and therefore frequency of sampling may impact results. The Pollinator Observations section (Line 107) should be expanded on by answering the above questions, providing rationale for temporal sampling decisions, and making sample sizes less ambiguous.
2. Lines 137-139 are confusing. The preceding sentences detail the methods used for determining landscape-level characteristics which led me to think that floral abundance was recorded at a similar scale. I later realized that it was recorded at the plot-level. Please add this detail to these lines.
3. Lines 196-199 provide results on pollinator visitation per flower. It is unclear what per flower means because the methods describe plot-level pollinator recording not flower-level. More description is needed in the methods section to make these results comprehensible.
4. Although some rationale is provided for sampling in the morning (Lines 109-111), it is unclear how it was determined that morning vs. afternoon sampling are equivalent.

Validity of the findings

1. Without knowing the exact protocol for sampling across space and time, it is impossible to assess the validity of the findings. Sample sizes need to be made less ambiguous in the methods as well as reported in tables/ figures (and/ or their captions). For example, it is unclear what each point represents in Figs 1 & 2. Does a single point represent a site average across plots? Or does a single point represent a plot average over multiple observation periods?
2. In addition, the validity of the findings are put into question by the inclusion of an outlier data point. The floral abundance panels in Fig. 1 and Fig. 2 show that there is an outlier point that may be affecting results. Further visual inspection of the data should be completed to confirm outlier status. It is advisable that if this point is an outlier, it should be removed and all analyses should be re-run. Changes to the results due to the exclusion of this outlier should be reported and subsequent interpretations updated to reflect these changes. The outlier site (School Sumran Wittaya) was only sampled at two plots which may be leading to an extreme result. There also may be biological reasons for excluding this sample (e.g., one of the plots contains a vine species with thousands of flowers that appear to be relatively unattractive to pollinators).
3. Based on the inclusion of the outlier, I have to wonder if the cut off for determining attractive flowers was inadequate. In addition to citing papers that have removed unattractive flowers from floral abundance counts before, it might be worth reviewing how attractiveness cut-offs have been determined in other papers. How do your results change if the threshold for attractiveness is changed?

Additional comments

This paper has the potential to spur much needed research on pollinator diversity in tropical cities. Unfortunately, a few major flaws prevent it from immediate publication. Some of the flaws are easily fixed by re-running analyses, however, without more information on how plots were sampled across time it is hard to judge whether the data are also plagued by poor experimental design.

---

## Round 0.2 · accepted · Accept

The authors clearly addressed all the comments and concerns, including those of the reviewer who recommended major revision, in the manuscript and in your reply letter. Congratulations.

#